# Trajectory Generation and Optimization Using the Mutual Learning and Adaptive Ant Colony Algorithm in Uneven Environments

**Feng Qu** [1], **Wentao Yu** [1,*], **Kui Xiao** [1], **Chaofan Liu** [1] **and Weirong Liu** [2]

[1] College of Computer and Information Engineering, Central South University of Forestry and Technology, Changsha 410004, China; qufeng0817@csuft.edu.cn (F.Q.); 20191100304@csuft.edu.cn (K.X.); cfliv@csuft.edu.cn (C.L.)

[2] School of Computer Science and Engineering, Central South University, Changsha 410083, China; frat@csu.edu.cn

\* Correspondence: wentaoyu@csuft.edu.cn; Tel.: +86-0731-134-6758-9376

**Abstract:** Aiming at the trajectory generation and optimization of mobile robots in complex and uneven environments, a hybrid scheme using mutual learning and adaptive ant colony optimization (MuL-ACO) is proposed in this paper. In order to describe the uneven environment with various obstacles, a 2D-H map is introduced in this paper. Then an adaptive ant colony algorithm based on simulated annealing (SA) is proposed to generate initial trajectories of mobile robots, where based on a de-temperature function of the simulated annealing algorithm, the pheromone volatilization factor is adaptively adjusted to accelerate the convergence of the algorithm. Moreover, the length factor, height factor, and smooth factor are considered in the comprehensive heuristic function of ACO to adapt to uneven environments. Finally, a mutual learning algorithm is designed to further smooth and shorten initial trajectories, in which different trajectory node sequences learn from each other to acquire the shortest trajectory sequence to optimize the trajectory. In order to verify the effectiveness of the proposed scheme, MuL-ACO is compared with several well-known and novel algorithms in terms of running time, trajectory length, height, and smoothness. The experimental results show that MuL-ACO can generate a collision-free trajectory with a high comprehensive quality in uneven environments.

**Keywords:** trajectory generation and optimization; mutual learning; adaptive ant colony algorithm; 2D-H map; de-temperature function





## 1. Introduction

Mobile robots are increasingly used in outdoor applications, such as search and rescue missions, planetary ground surveys, and national defense security [1] where robots usually have to face complex environments with an uneven terrain [2–4]. In such an environment, the planned trajectory usually includes path segments with rapidly changing height and multiple sharp turns. When the mobile robot passes through this section of the path, it will consume more energy [5], which should be avoided. Therefore, this paper focuses on trajectory generation and optimization to quickly find a short and smooth trajectory in uneven environments.

In an uneven environment, it is obvious that ordinary 2D grid maps cannot reasonably model the environment, so it is essential to build a map model that can better describe the uneven environment. Generally, 3D modeling and 2.5D elevation maps are used in this circumstance. In [6–8], 3D radar for 3D modeling was used which simulates the complex uneven outdoor environment including stairs, slopes, etc. This method can accurately and comprehensively grasp the specific information of the surrounding environment [9], but it occupies a large amount of computer running memory and has poor timeliness, especially

when the size of the environment increases. Therefore, scholars have tried to find simpler and more applicable modeling methods. In [10–12], the authors tried to reduce the cost of 3D radar modeling, using RGB-D (Red Green Blue-Depth) sensors to store the height information and coordinates of the environment in the same grid to obtain a discrete 3D grid map, namely a 2.5D elevation map. This paper proposes a 2D-H grid map based on a 2.5D elevation map, which maps the height information of the environment to a 2D plane. Compared with the 2.5D elevation map, the 2D-H method further reduces the number of calculations and saves memory space, which not only meets the needs of timeliness but is also practical enough.

The goal of trajectory planning is to find a trajectory for a robot from a starting point to an ending point without colliding with obstacles and meeting other constraints, such as the trajectory length and smoothness. In recent years, trajectory planning has attracted much attention in the field of robotics. Traditional methods, such as simulated annealing (SA) [13] and the artificial potential field method (APF) [14], are widely used because of their ease of understanding and implementation. However, [15] these methods may easily fall into a local optimal solution and cannot reach the end point when a complex environment is encountered.

When dealing with trajectory planning in a complex dynamic environment, intelligent bionic algorithms often play an important role [16], including genetic algorithms (GA) [17], neural networks [18], particle swarm optimization (PSO) [19], and the ant colony algorithm (ACO) [20]. The GA algorithm has strong uncertainty in the planned trajectory due to its large number of parameters and the randomness of the initial population [21]. In the PSO algorithm, [22] the initial particles are randomly selected, which leads to a variety of different planning results. [23]. Neural network algorithms spend a lot of time on pre-training and adjusting parameters. Not only is the number of calculations significant, but also the interpretability is not ideal. In an uneven environment with obstacles, the uncertainty of the environment, such as the distribution of obstacles and the complexity of the environment, brings difficulties to the trajectory planning of mobile robots. Therefore, the required path planning algorithm needs a strong search ability and good stability.

ACO is a probabilistic technique to solve computational problems which is robust and easy to combine with other methods [24,25]. In recent years, the ant colony algorithm has been widely used in transportation, logistics distribution, network analysis, and other fields [26]. Nevertheless, the traditional ant colony algorithm has the defects of a low search efficiency and a slow convergence speed, and it is also easy to fall into local extremes.

Researchers have put forward various improved methods to optimize the search ability of the ant colony algorithm. Li et al. [27] proposed an improved ant colony algorithm with multiple heuristics (MH-ACO), which is better reflected in the global search ability and convergence. However, the parameter setting is complex, which brings randomness to the experiment. Akka et al. [28] used a stimulating probability to help the ants choose the next grid and employed new heuristic information based on the principle of unlimited step length to expand the field of view and improve the visibility accuracy. The improved algorithm speeds up the convergence speed and expands the search space, but the safety of the trajectory cannot be guaranteed. In addition, Ning et al. [29] designed a new pheromone smoothing mechanism to enhance the search performance of the algorithm. When the search process of the ant colony algorithm is close to stagnation, the pheromone matrix is reinitialized to increase the diversity of the connections at the expense of a large time cost.

Furthermore, the improved ant colony algorithm is combined with some two-stage trajectory planning methods. Chen et al. [30] proposed a fast two-stage ant colony algorithm based on the odor diffusion principle, including two stages of preprocessing and trajectory planning, which accelerated the convergence speed but did not consider the trajectory optimization. Yang et al. [31] proposed a multi-factor improved ant colony algorithm (MF-ACO) to solve the problem related to the fact that the trajectory planning algorithm of mobile robots cannot cope with the complex actual environment. The maximum and minimum ant strategy was adopted to avoid local optima. Then the dynamic tangent point

adjustment method was used to smooth the path to further improve the quality of the trajectory, but the smoothness needed to be further improved.

Although the above-mentioned improved algorithm attempts to improve the search performance of ACO and speed up the convergence speed, it does not consider the height information in the environment, which is different from ordinary raster maps, and the convergence performance of ACO can be further improved. For trajectory planning and optimization in complex and uneven environments, this paper proposes a trajectory generation and optimization method based on mutual learning and adaptive ant colony optimization (MuL-ACO), which can make robots safely and quickly plan a short and smooth trajectory in uneven environments. In this method, the global trajectory planning of mobile robots is divided into two consecutive parts: the initial trajectory generation and trajectory optimization. Firstly, the improved adaptive ant colony algorithm further accelerates the convergence of ACO to quickly generate the initial trajectory. The initial planned trajectory may contain redundant points and inflection points, which results in high memory consumption and poor trajectory quality. Then, a trajectory optimization algorithm based on mutual learning is proposed to further optimize the length and smoothness of the initial trajectory.

The main contributions of this paper are as follows.

1. The 2D-H raster map is proposed to simulate the uneven outdoor environment. The height information in the three-dimensional environment is stored in the 2D plane;
2. A hybrid scheme using mutual learning and adaptive ant colony optimization is proposed in this paper. The global robot trajectory planning problem is divided into two consecutive parts: the trajectory generation part and the trajectory optimization part;
3. An improved adaptive ant colony algorithm is proposed to generate the initial trajectory. Considering the height information of the map, a comprehensive heuristic function including length, height, and smoothness is designed. Then, a new pheromone adaptive update strategy is proposed through an improved simulated annealing function to speed up the convergence of the algorithm.
4. A new trajectory optimization algorithm based on mutual learning is proposed to optimize the generated initial trajectory. Firstly, feature ablation experiments are carried out for each turning point to obtain the safety feature sequence of each turning point. Then, each point learns from other points to gradually eliminate the points that do not affect the trajectory safety to optimize the trajectory length and smoothness. Finally, the shortest sequence of key points affecting trajectory safety is obtained. Therefore, the algorithm optimizes the final trajectory in terms of smoothness, length, and stability.

This research is structured as follows: Section 2 describes the environment and problems. Section 3 illustrates the improved adaptive ant colony algorithm. Section 4 describes the framework and the process of the proposed trajectory optimization algorithm based on mutual learning in detail. Section 5 presents the steps and flowcharts of a hybrid scheme using mutual learning and adaptive ant colony optimization. Section 6 discusses the results of the simulation experiment. Finally, Section 7 concludes the paper.

## 2. Environment Description and Problem Formulation

### 2.1. Environment Description

In an uneven environment, the working environment of a mobile robot is a 2D-H grid map. The grid of the map is divided into grids occupied by obstacles, grids with height information, and free grids.

In the 2D-H map, the obstacle grid is considered impassable, the free grid is considered passable, and the height grid affects the quality of trajectories planned by robots. The height grid is modeled by a normalized method, as shown in Equation (1). $H(i)$ is the color intensity value of the $i$-th height grid cell, and the color becomes darker as the value of $H(i)$

increases, $H(i) \in [0, 1]$. $h$ is a height function when $h(i) > 0$ and $H$ represents the intensity value of green, otherwise $H$ represents the intensity value of blue.

$$H(i) = \begin{cases} \frac{\max(h) - h(i)}{\max(h)}, & h(i) > 0 \\ \frac{\min(h) - h(i)}{\min(h)}, & h(i) < 0 \end{cases} \tag{1}$$

Figure 1 shows the modelling process of the 2D-H map. The uneven environment here is simulated with the peaks function, a probability density function of a binary Gaussian distribution. Researchers can design other functions to simulate an uneven environment according to different requirements. Figure 1a shows the three-dimensional model of the mountain function and the contour mapping on the 2D plane. Figure 1b shows the 2D-H map generated by the peak function and obstacles, where black grid cells represent obstacles, blue cells represent concave areas, and green cells represent raised areas. Here the height information in the map and the location of static obstacles are known.

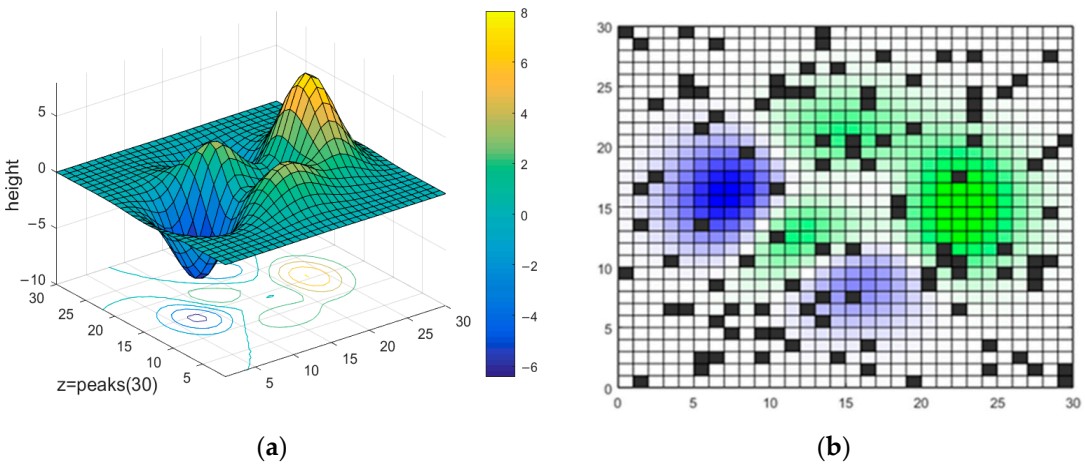

**(a)**

**(b)**

**Figure 1.** The modelling process of the 2D-H map. (**a**) The 3D map of peak function. (**b**) The 2D-H map generated by the peak function and obstacles.

### 2.2. Problem Formulation

Based on the 2D-H map, the problem of trajectory generation and optimization is described as follows. Given a starting point and an ending point, a robot is expected to plan an initial trajectory between them. In the trajectory generation stage, the optimization is to minimize the length, height difference, and number of turns of the trajectory. Then, in the trajectory optimization stage, redundant nodes of the initial trajectory are further reduced to obtain a shorter and smoother trajectory.

The robot can only move one grid distance in a time step, the side length of one grid is set to 1 m, and the height threshold is set to (−1 m, 1 m) to limit the movement of the robot. The mobile robot is regarded as a mass point and moves at a fixed speed. As shown in Figure 2, there are eight moving directions of the robot, which point from the center of the current grid to the center of the adjacent eight grids. Figure 2b shows a trajectory of the robot moving from the starting point to the target point. The planned trajectory is limited by the height threshold, so the darkest blue and green grids are bypassed.

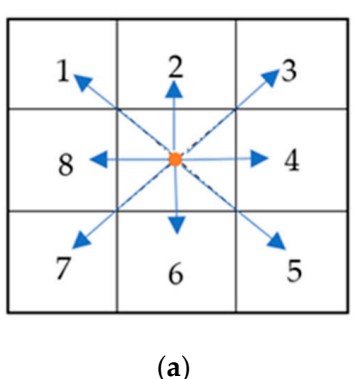
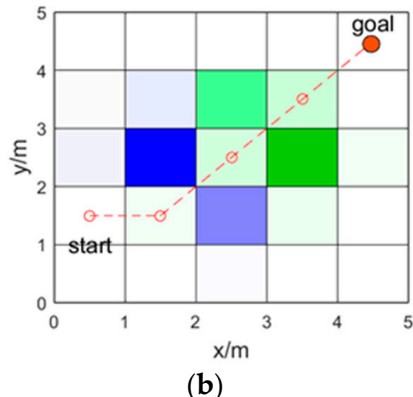

(**a**)                                    (**b**)

**Figure 2.** (**a**) The moving direction of the robot; (**b**) The planned trajectory limited by the height threshold.

## 3. Improved Adaptive Ant Colony Algorithm

### 3.1. Basic Ant Colony Optimization

Ant colony optimization is a metaheuristic algorithm inspired by the behavior of ants in nature. It imitates the foraging behavior of ant colonies to find the optimal trajectory in an unknown environment. Two factors determine the next step of ants, which are heuristic information and pheromones. Heuristic information is obtained from the surrounding environment, and pheromones are the directional information emitted by the group. In the process of food searching, the ants release pheromones on the trajectory they walked by, which will attract other ants. As the number of ants on the same trajectory increases, pheromones are gradually accumulated, and more ants are attracted to the trajectory. According to the surrounding environment and experience, the transition probability of ants can be calculated by Equation (2).

$$
p_{i,j}^k(t) = \begin{cases} \dfrac{[\tau_{i,j}(t)]^a [\eta_{i,j}^k(t)]^\beta}{\sum\limits_{s \in A_i} [\tau_{i,j}(t)]^a [\eta_{i,j}^k(t)]^\beta}, & j \in A_i \\ 0, & j \notin A_i \end{cases} \tag{2}
$$

where $p_{i,j}^k(t)$ is the transition probability of the $k$-th ant from point $i$ to point $j$ at the $t$-th iteration, $\tau_{i,j}(t)$ is the pheromone concentration from point $i$ to point $j$, $\eta_{i,j}^k(t)$ is the heuristic function of the $k$-th ant from point $i$ to point $j$, $A_i$ is a set of feasible points adjacent to $j$, and $\alpha$ and $\beta$ are weights that control the relative importance of $\tau_{i,j}$ and $\eta_{i,j}$. If the proportion of $\alpha$ is larger, then ants tend to choose the trajectory that most of the previous ants have walked. If the proportion of $\beta$ is larger, ants tend to choose the trajectory to traverse based on the heuristic information in the environment. Heuristic function and pheromone concentration directly affect the behavior of ants searching for trajectories. The heuristic function $\eta_{i,j}^k(t)$ can be calculated by Equation (3), where $d_{jg}$ is the Euclidean distance between the next node $j$ and the end point $g$.

$$
\eta^k{}_{i,j}(t) = \frac{1}{d_{jg}} \tag{3}
$$

When all the ants complete the trajectory search, the pheromones will accumulate on the passing trajectory. The pheromone $\tau_{i,j}(t)$ is updated once per iteration, as shown in Equation (4).

$$
\tau_{i,j}(t+1) = (1-\rho)\tau_{i,j}(t) + \sum_{k=1}^m \Delta\tau_{i,j}^k(t) \tag{4}
$$

$$
\Delta\tau_{i,j}^k(t) = \begin{cases} Q/L_k, & \textit{if ant } k \textit{ visits } i \textit{ and } j \\ 0, & \textit{otherwise} \end{cases} \tag{5}
$$

Here $\Delta\tau_{i,j}^{k}(t)$ is the pheromone increment left by ant $k$ after passing the trajectory between $i$ and $j$, at the $t$-th iteration. $\rho$ is the pheromone volatilization factor, which adjusts the accumulation speed of the pheromone. Moreover, $m$ is the total number of ants. Generally, $\Delta\tau_{i,j}(0) = C$, $C$ is the initial pheromone constant. $Q$ is a constant, and $L_k$ is the total length of the trajectory passed by ant $k$ in this iteration, which is inversely proportional to $\Delta\tau_{i,j}^{k}(t)$.

### 3.2. Improved Heuristic Function

To reduce the blindness of the ants in the early stage and to find a short and smooth trajectory in an uneven environment, the heuristic function is redesigned. The smoothness and height factors are added to the heuristic function, as shown in Equation (6). The comprehensive heuristic function includes length $d$, a smoothing function s, and a height function $h$.

$$\eta_{i,j}(t) = \lambda d(i,j,g) + \gamma h(i,j) + \psi s^{k}{}_{i,j}(t) \tag{6}$$

In the heuristic function, the distance function $d(i,j,g)$, height function $h(i,j)$, and smoothing function $s_{i,j}^{k}(t)$ are defined as follows. $\lambda, \gamma, \psi$ are weight parameters and they depend on the application.

$$d(i,j,\text{g}) = \frac{d_{\max(j,g)} - d_{j,g}}{d_{\max(j,g)} - d_{\min(j,g)} + 0.01} \times \omega \tag{7}$$

where $d(i,j,g)$ is the corrected distance from a certain adjacent grid cell $j$ of the current grid cell $i$ to the target grid cell. $d(i,j,g)$ is designed to enlarge the distance gap between adjacent grid cells. $d_{\max}$ and $d_{\min}$ are the maximum and minimum distances between adjacent grid cells and the end point. $w$ is a correction parameter, and 0.01 avoids the situation where the denominator is 0.

$$h(i,j) = \begin{cases} \frac{h_{\max} - |h(i) - h(j)|}{h_{\max} - h_{\min} + 0.01} \times \omega, & |h(i) - h(j)| \leq h_c \\ \infty, & |h(i) - h(j)| > h_c \end{cases} \tag{8}$$

where $h(i,j)$ is the corrected height from the current grid cell $i$ to a certain neighboring grid cell $j$. $h(i,j)$ is designed to guide ants to visit flatter grid cells. $h_c$ is the height constraint value of the robot, which should meet the limit of the height threshold. $h_{\max}$ and $h_{\min}$ are the maximum and minimum height differences between adjacent grid cells and the current grid cell. $w$ is a correction parameter, and 0.01 avoids the situation where the denominator is 0.

$$s_{i,j}^{k}(t) = \begin{cases} u, & dir_{a,i}^{k}(t) = dir_{i,j}^{k}(t) \\ 0.1 * u, & otherwise \end{cases} \tag{9}$$

where $s_{i,j}^{k}(t)$ is the smoothness of the trajectory passed by the $k$-th ant in the $t$-th iteration. Generally, ants can move in eight directions adjacent to the current grid. Suppose an ant reaches the current point $i$ from $a$, and then goes to the next point $j$. If the moving direction of the previous step $dir_{a,i}^{k}$ is consistent with the moving direction of $dir_{i,j}^{k}$, the reward $u$ will be given, and the value of $u$ is set to 5 in this paper.

### 3.3. SA-Based Adaptive Adjustment Strategy of Pheromones

In the traditional ant colony algorithm, the pheromone is often a fixed value and will not automatically adjust with the iteration situation. Ants search blindly in the early stage due to the low concentration of pheromones. In the later stage, pheromones will accumulate in a large amount, which will reduce the diversity of trajectory selection. Therefore, the fixed increase in the pheromones cannot meet the requirements of search efficiency and will slow down the convergence speed.

Based on the temperature update function of the simulated annealing algorithm, this paper proposes a new annealing strategy to dynamically update the pheromone volatilization factor. The temperature $T$ is an important control parameter in the simulated annealing algorithm, which determines the annealing direction and the running speed of the algorithm. As the iteration makes the de-temperature function have linear characteristics, the running speed of the de-temperature function is further improved. Moreover, a dynamic pheromone volatilization factor $\rho$ is designed to realize the adaptive update of pheromone. $\rho$ is set by Equations (10) and (11).

$$\rho = l - \exp(-(\mu/T_i)) \tag{10}$$

$$T(i) = \begin{cases} T_{\text{end}} + \frac{(T_{\text{start}} - T_{\text{end}}) * (I_{\text{max}} - I_i)}{I_{\text{max}}}, & I_i \leq \frac{I_{\text{max}}}{2} \\ T_{\text{end}} + \frac{(T_{\text{start}} - T_{\text{end}}) * (I_{\text{max}} - (I_{\text{max}} - I_i))}{I_{\text{max}}}, & I_i > \frac{I_{\text{max}}}{2} \end{cases} \tag{11}$$

where $\mu$ is a constant and $T_i$ is the current temperature, which is calculated by the de-temperature function $T(i)$. $I_{\text{max}}$ is the number of maximum iterations. $T_{start}$ is the initial temperature, and $T_{end}$ is the final temperature. As the number of iterations increases, the de-temperature function linearly decreases and then rises to further increase the running speed of the de-temperature function. To achieve better experimental results, the value of parameter $\mu$ is discussed in this paper. Other parameters are set to: $T_{start} = 100$, $T_{end} = 0.1$, $I_{\text{max}} = 50$.

In the previous work, many experiments were performed to determine the value of $\mu$, and finally three possible and suitable values were selected and are discussed in this paper. The values of $\mu$ were set to 33, 50, and 90, respectively, to calculate $\rho$ according to Equation (10). The adaptive change curve of $\rho$ is shown in Figure 3. In the early stage, the volatilization factor $\rho$ is small, which is conducive to accumulating pheromones and improving the directionality of the ants' search. In the mid-term, $\rho$ becomes larger to speed up the iteration speed of the algorithm and to avoid falling into the local optimum. In the later stage, $\rho$ becomes small to accelerate convergence.

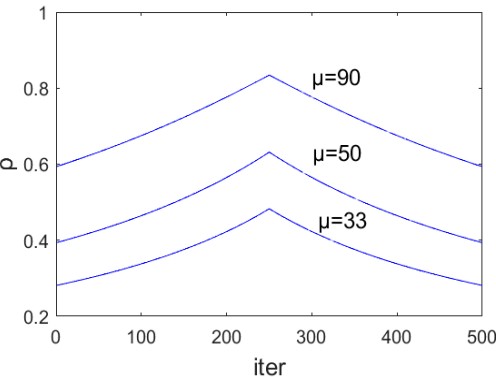

**Figure 3.** Adaptive change of the pheromone volatilization factor.

To further test the influence of the value of $\mu$ on the convergence of the improved adaptive ant colony algorithm (IAACA) proposed in this paper, three benchmark functions were applied for verification. Other swarm intelligence algorithms including ACO, PSO, Grey Wolf Optimizer (GWO) [32] were chosen to compare with IAACA in terms of convergence performance. Table 1 presents the details about the different types of benchmark functions. The three benchmark functions had a minimum value of 0, the particle dimension was set to 10, and the value range of x and y was (−5.12, 5.12). Sphere is a unimodal function, while Rastrigin and Ackley are multimodal functions with many local values and difficult to solve. These three benchmark functions were suitable for testing the convergence performance of the algorithm.

**Table 1.** Description of benchmark functions.

| Functions | Function Expressions | Range | $D_{im}$ | $f_{min}$ |
|---|---|---|---|---|
| Sphere | $F_1 = \sum\limits_{i=1}^{D} x_i^2$ | $[-5.12, 5.12]$ | 10 | 0 |
| Rastrigin | $F_2 = \sum\limits_{i=1}^{D} \left[ x_i^2 - 10\cos(2\pi x_i) + 10 \right]$ | $[-5.12, 5.12]$ | 10 | 0 |
| Ackley | $F_3 = -20\exp\left(-0.02\sqrt{\frac{1}{D}\sum\limits_{i=1}^{D} x_i}\right) - \exp\left(\frac{1}{D}\sum\limits_{i=1}^{D}\cos(2\pi x_i)\right) + 20 + e$ | $[-5.12, 5.12]$ | 10 | 0 |

The convergence curves of PSO, GWO, ACO, and IAACA ($\mu = 33$, $\mu = 50$, $\mu = 90$) corresponding to the three benchmark functions are shown in Figure 4. It can be seen from Figure 4 that the convergence performance of IAACA was the best when $\mu = 33$. As shown in Figure 4a, the convergence performance of IAACA ($\mu = 33$) was better than the other three algorithms. Although GWO converged rapidly downward, it still did not converge in the end. In Figure 4b,c, IAACA ($\mu = 33$) and GWO converged at around 50 iterations, and the convergence performance was better than the other two algorithms. Figure 4b,c showed that IAACA ($\mu = 33$) and GWO had a better convergence performance, but IAACA ($\mu = 33$) was more stable from Figure 4a.

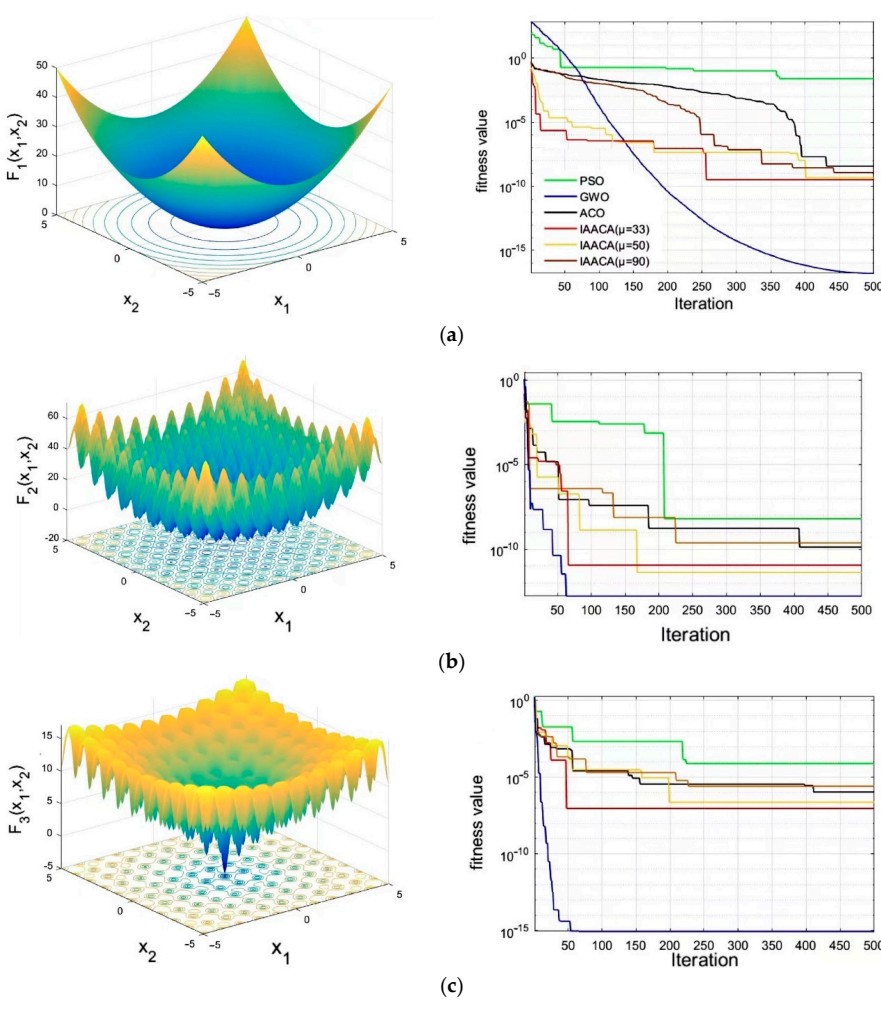

**Figure 4.** Convergence curves for PSO, GWO, ACO, and IAACA ($\mu = 33$, $\mu = 50$, $\mu = 90$) on different benchmark functions. (**a**) Convergence performance of swarm intelligence algorithms on the Sphere function. (**b**) Convergence performance of swarm intelligence algorithms on the Rastrigin function. (**c**) Convergence performance of swarm intelligence algorithms on the Ackley function.

## 4. Mutual Learning-Based Trajectory Optimization Algorithm

To reduce the redundant nodes of the initial trajectory generated by the adaptive ant colony algorithm to further optimize the length and smoothness of the trajectory, a trajectory optimization algorithm based on mutual learning is proposed. Firstly, feature ablation experiments were carried out for each turning point to obtain the safety feature sequence of each turning point. Then, each point learns from other points to gradually eliminate the points that do not affect the trajectory safety to optimize the trajectory length and smoothness. Finally, the shortest sequence of key points affecting trajectory safety is obtained. The proposed algorithm achieved a smooth trajectory and minimized the length of the trajectory. At the same time, the wear of the robot's steering to follow the planned trajectory was reduced.

For instance, as shown in Figure 5, the feasible initial trajectory $L_{old}$: $S \rightarrow N_1 \rightarrow \ldots \rightarrow N_8 \rightarrow \ldots \rightarrow E$ is usually not the best trajectory. After the trajectory optimization algorithm based on mutual learning is optimized once, the trajectory $L_1$ can reach the end point without passing through the $N_1$ point. When the optimization is completed, the final trajectory $L_{best}$ can even reach the end point directly through the $N_8$ point. Consequently, $L_{old}$ is optimized to $L_{best}$: $S \rightarrow N_8 \rightarrow E$, which optimizes the length and smoothness of the initial trajectory. The algorithm is described as follows.

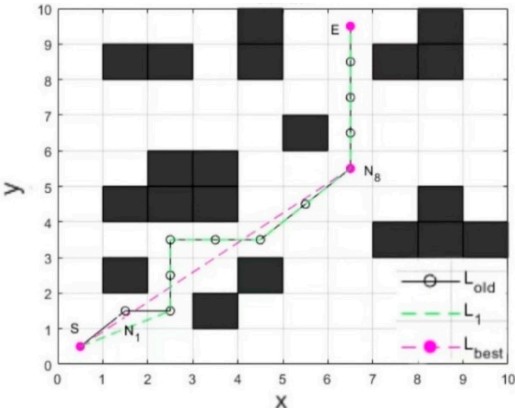

**Figure 5.** The initial trajectory and the trajectory optimized by mutual learning.

The initial trajectory $L_{old}$, including the starting point $S$ and the ending point $E$, can be represented by a set of all turning points $N = \{S, N_1, N_2, \ldots, N_i, \ldots, N_n, E\}$, $i \in [1, n]$, and the coordinates of the point set are represented by Equation (12).

$$R = \{(x_0, y_0), (x_1, y_1), (x_2, y_2), \ldots, (x_i, y_i), \ldots, (x_n, y_n), (x_{n+1}, y_{n+1})\} \tag{12}$$

To learn the collision characteristic of each turning point, it is assumed that there are $n$ initial individuals $P_i$ represented by the characteristic matrix (13). Then each individual subjected to a characteristic ablation experiment, and the characteristic zero point is set by Equation (14).

$$P_i = [N_0, N_1, N_2, \ldots, N_i, \ldots, N_n, N_{n+1}] \tag{13}$$

$$N_i = \begin{cases} 0, & N_i \text{ is ablated} \\ 1, & \text{otherwise} \end{cases} \tag{14}$$

A reward and punishment step are added to determine whether the $i$-th individual $P_i$ reaches the target directly from the starting point without collision. The cost function is $L_i(P_i)$ calculated by Equation (15), $L_{old}$ is the length of the initial trajectory, and $L_i$ is the trajectory length of the current individual.

$$L_i(P_i) = \begin{cases} L_i, & \text{if } P_i \text{ no collision} \\ L_{old}, & \text{otherwise} \end{cases} \tag{15}$$

$$L_i = \sum_{k=0}^{n+1} d_{N_k,N_{k+1}} = d_{N_0,N_1} + d_{N_1,N_2} + \ldots + d_{N_{k-1},N_k} + d_{N_k,N_{k+1}} + \ldots + d_{N_n,N_{n+1}}, if(N_r = 0)$$
$$= d_{N_0,N_1} + d_{N_1,N_2} + \ldots + d_{N_{r-1},0} + d_{0,N_{r+1}} + \ldots + d_{N_n,N_{n+1}}$$
$$= d_{N_0,N_1} + d_{N_1,N_2} + \ldots + d_{N_{r-1},N_{r+1}} + \ldots + d_{N_n,N_{n+1}}$$

(16)

$$d_{N_k,N_{k+1}} = \sqrt{(x_k - x_{k+1})^2 + (y_k - y_{k+1})^2}$$

(17)

The current individual $P_i$ learns from each other in turn with other individuals $P_j$, and learns the collision characteristics of each other's nodes to obtain the best individual. Suppose $P_1$ is the current individual, the mutual learning process is shown in Figure 6.

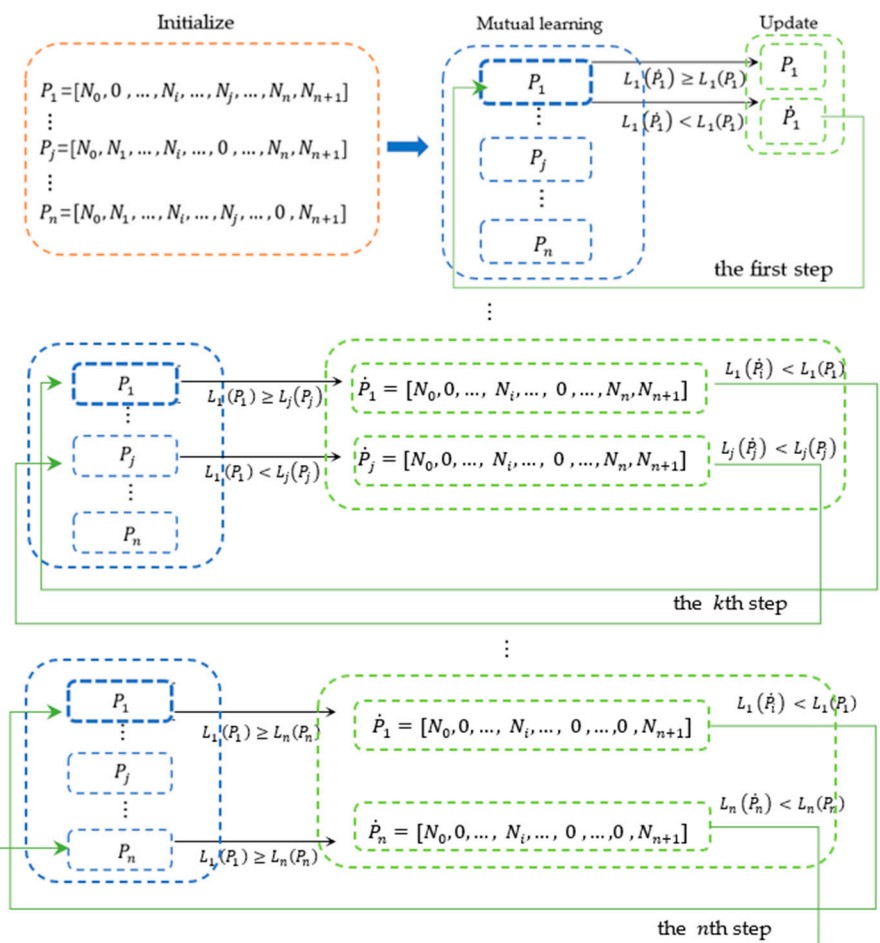

**Figure 6.** The mutual learning process of $P_1$.

In the first stage of initialization, each turning point is subjected to an ablation experiment to obtain the feature and feature sequence of the point, which can be obtained by Equation (14). In the second stage of mutual learning, individuals start from $P_1$ and learn from each other in turn. The learning method is to compare the value of the cost function, and individuals with high values learn from individuals with low values. In the third stage of the individual update after mutual learning, if the current individual penalty value $L_i$ does not increase, the old individual $P_i$ is updated to the new individual $\dot{P}_i$ according to the cost function, otherwise, it is not updated. The individual is updated by Equation (18).

$$\hat{P}_i = \begin{cases} \dot{P}_i, & if\ \hat{L}_i(\dot{P}_i) < L_i(P_i) \\ P_i, & otherwise \end{cases}$$

(18)

The ultimate goal of the mutual learning trajectory optimization algorithm is to generate an optimized path with fewer turns $D_{best}$ and a shorter length $L_{best}$.

$$L_{best} = \min\{\hat{L}_1, \hat{L}_2, \cdots, \hat{L}_i, \cdots, \hat{L}_n\} \tag{19}$$

$$D_{best} = \text{length }\{P_{best} \neq 0\} \tag{20}$$

Algorithm 1. The pseudo code for MLTO.

| **Algorithm 1.** Mutual learning-based Trajectory Optimization Algorithm |
| --- |
| 1: input turning point set $N$ |
| 2: initialize point set $N$ as feature individual $P_i$ by (13)(14) |
| 3: calculate the reward and punishment function $L_i$ by (15)(16)(17) |
| 4: for    $i = 1$ to $n$ do |
| 5:      for    $j = 1$ to $n$ do |
| 6:            if    $L_j$ is not more than $L_i$ then |
| 7:              $P_i$ learns feature zero through $P_j$, and calculate $\hat{L}_i$ |
| 8:              $P_i$ will be updated to $\hat{P}_i$ by (18) |
| 9:            else |
| 10:               $P_j$ learns feature zero through $P_i$, and calculate $\hat{L}_j$ |
| 11:               $P_j$ will be updated to $\hat{P}_j$ by (18) |
| 12:          end    if |
| 13:      end    for |
| 14: end    for |
| 15: if    $L_{best}$ is equal to the initial trajectory $L$ then |
| 16:      the optimal trajectory is initial trajectory $L$, number of turns is $D$ |
| 17: else |
| 18:    calculate the length of the optimal trajectory and the number of turns by (19)(20) |
| 19: end if |
| 20: output the number of turns: $D_{best}$ |
| 21: output the shortest trajectory: $L_{best}$ |

## 5. A Hybrid Scheme Using MuL-ACO for Trajectory Generation and Optimization

For the robot to effectively find a short and smooth trajectory while avoiding crossing steep areas, a hybrid scheme using mutual learning and adaptive ant colony, namely, MuL-ACO, is presented in this work. Figure 7 shows the flow chart of the scheme.

Step 1:    Establish a 2D-H grid map based on the uneven environment and initialize the parameters. Set the starting point $S$, the end point $E$, the number of ants $k$, the maximum number of iterations $I_{max}$, pheromone heuristic factor $\alpha$, expected heuristic factor $\beta$, pheromone intensity coefficient $Q$, and the initial pheromone $\tau(0)$.

Step 2:    Trajectory selection. Calculate the heuristic function by Equation (6). The ant is placed at the starting point and the probability of transferring to the next node is calculated by Equation (2). All trajectory nodes from the starting point to the current point are stored in the Tabu list.

Step 3:    Determine whether all the ants have completed the trajectory search in this generation. If it is, go to step 4, otherwise, return to step 2.

Step 4:    Record the nodes of the trajectory walked by all ants and find the optimal trajectory in this iteration.

Step 5:    Adjust the pheromone volatilization factor $\rho$ adaptively, according to Equations (10) and (11). Update the pheromone by Equation (4). Re-zero the taboo table.

Step 6:    Determine whether the maximum number of iterations $I_{max}$ is reached. If it is, go to step 7, otherwise, return to step 2.

Step 7:    Initial trajectory generation. Obtain the optimal trajectory node and total height.

Step 8: Trajectory optimization. The trajectory optimization algorithm based on mutual learning, such as Algorithm 1, is used to optimize the initial trajectory. Calculate the optimal trajectory including length, height, and number of turns.

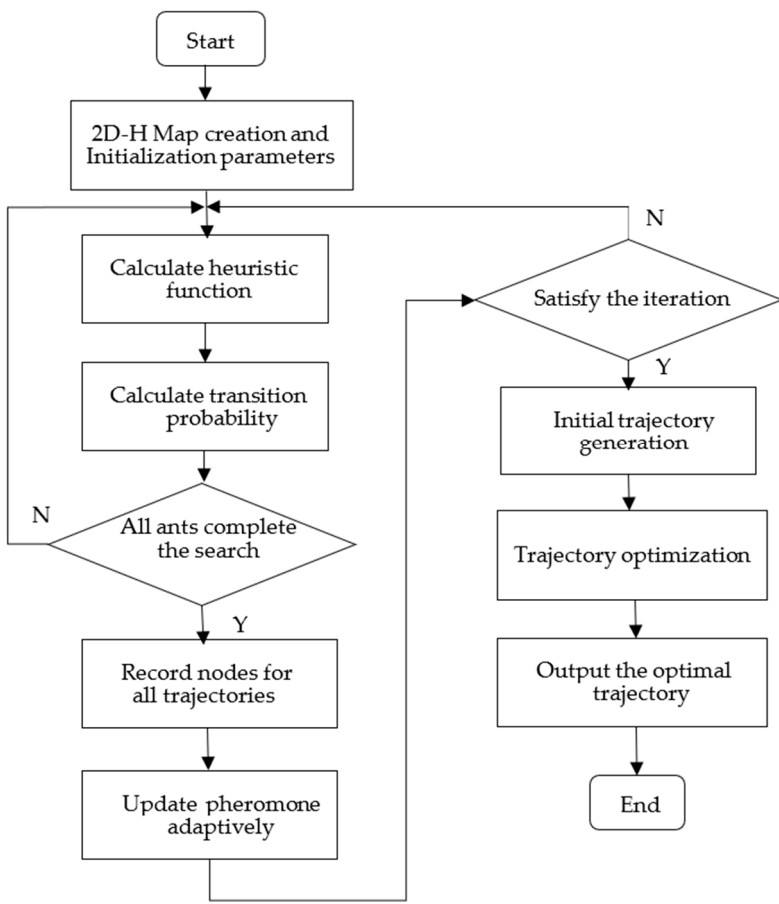

**Figure 7.** Flow chart of the MuL-ACO.

## 6. Experiment

In this section, four sets of simulation experiments were conducted to evaluate the performance of the MuL-ACO scheme. In the first set of experiments, the algorithm proposed in this paper was compared with other intelligent algorithms, which are the well-known GA and the novel sparrow search algorithm (SSA) [33]. In the second set of experiments, the algorithm proposed in this paper was compared with the traditional ACO and other improved ACO, which are MH-ACO and MF-ACO. Two groups of experiments were carried out on maps with different sizes and different numbers of obstacles. The third set of experiments was set with different starting points and ending points on a map, and compared with the proposed algorithm with ACO, MH-ACO, and MF-ACO. To further verify the effectiveness of the improved algorithm, the fourth set of experiments was simulated in a dynamic environment. Furthermore, the height threshold of uneven environments was set to (−1 m, 1 m) to constrain the robot's trajectory search. The initial trajectory is given on each map to show the process of trajectory generation and optimization.

To build the environment map of the mobile robot, 2D-H grid maps with different sizes, obstacles, and terrains were modeled by the MATLAB simulation platform, where the blue areas are low terrain areas, and the green areas are high terrain areas. Obstacles were randomly placed on the map and start and end points were set. All experiments were performed on the same PC to obtain an unbiased comparison of CPU time. Parameters of MuL-ACO were set as the following: $k = 50$, $I_{max} = 50$, $\alpha = 3$, $\beta = 6$, $C = 10$, $Q = 100$, $u = 5$, $w = 8$, $h_c = 1$, $\tau(0) = 20$, $\mu = 33$, $\lambda = \gamma = \psi = 1$.

### 6.1. Simulation Experiment A

In this experiment, six maps of 20 × 20 m were selected for simulation, which differed in the number of obstacles and the shapes of obstacles. Figure 8 shows the optimal trajectories planned by GA, SSA, and MuL-ACO. To obtain more specific performance indicators, including length, number of iterations, smoothness, height difference, and running time, the experiment was performed 30 times to obtain the average value. Table 2 summarizes the qualitative comparison of the performance of MuL-ACO, GA, and SSA.

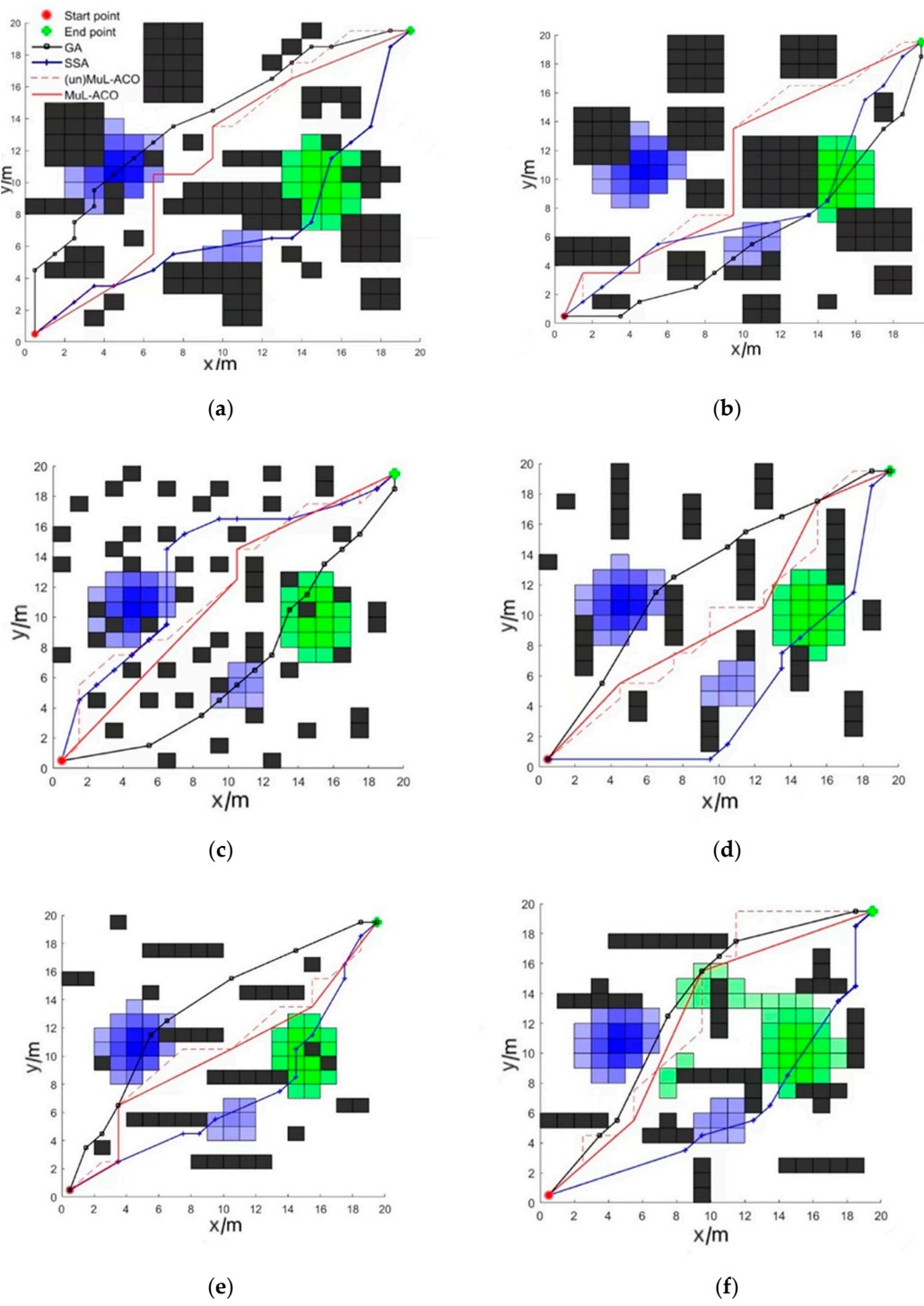

**Figure 8.** (**a**–**f**) Optimal trajectories generated by GA, SSA, and MuL-ACO on different 20 × 20 maps.

**Table 2.** Simulation results of GA, SSA, and MuL-ACO.

| Name | Map NO. | Length | Iterations | Turns | Height Difference | Time (s) |
|---|---|---|---|---|---|---|
| GA | a | 28.7 | 32 | 10 | 5.6 | 9.2 |
| | b | 29.7 | 38 | 9 | 5.4 | 12.5 |
| | c | 28.5 | 37 | 8 | 4.2 | 10.7 |
| | d | 28.1 | 31 | 5 | 5.3 | 10.4 |
| | e | 28.6 | 39 | 7 | 6.7 | 12.6 |
| | f | 29.0 | 34 | 6 | 4.1 | 9.5 |
| SSA | a | 29.6 | 28 | 9 | 4.3 | 0.7 |
| | b | 29.9 | 25 | 9 | 5.7 | 0.9 |
| | c | 30.7 | 26 | 7 | 3.1 | 0.5 |
| | d | 31.4 | 37 | 6 | 2.8 | 0.8 |
| | e | 28.9 | 21 | 10 | 7.2 | 0.6 |
| | f | 29.3 | 23 | 6 | 7.5 | 0.6 |
| MuL-ACO | a | 32.4 | 7 | 5 | 0.9 | 1.4 |
| | b | 29.8 | 6 | 6 | 1.2 | 1.3 |
| | c | 28.0 | 8 | 2 | 0.3 | 1.5 |
| | d | 27.9 | 9 | 3 | 0.5 | 1.7 |
| | e | 28.7 | 6 | 3 | 0.4 | 1.3 |
| | f | 28.2 | 7 | 2 | 2.1 | 1.6 |

In Figure 8a–f, the trajectory planning based on MuL-ACO tended to bypass steep or concave areas and generated a comprehensive optimal target trajectory with good safety, fewer turns, and a shorter length. The main reason is that MuL-ACO introduces the height factor in environments into the heuristic function, and the trajectory optimization algorithm based on mutual learning further optimizes the trajectories. In contrast, the GA and SSA algorithms do not actively bypass steep or concave areas when generating trajectories, so mobile robots will incur certain losses when following trajectories and the quality of trajectories is average. In the case of a small difference in trajectory length, as shown in Figure 8a,b,e, MuL-ACO can find a flatter and safer trajectory through the trajectory optimization algorithm based on mutual learning. In Figure 8c,d,f, the trajectories generated by MuL-ACO were shorter, smoother, and safer than GA and SSA.

As shown in Table 2, compared with GA and SSA, the hybrid scheme of MuL-ACO proposed in this paper had a large improvement in the convergence performance, which was about 70% to 85%. The reason is that the SA-based pheromone adaptive adjustment strategy accelerated the convergence of the algorithm. In terms of the height difference of the trajectory, the trajectory planned by MuL-ACO tended to avoid passing through steep uneven areas, so the total height difference of the trajectory was lower than GA and SSA. Compared with the other two algorithms, the final trajectory generated by the mutual learning-based trajectory optimization algorithm was better in terms of length and smoothness. In addition, it can be seen from Table 2 that MuL-ACO had a shorter running time when planning the trajectory.

*6.2. Simulation Experiment B*

In this experiment, six maps of 30 × 30 m were selected for simulation, which differed in the number of obstacles and the shapes of obstacles. Start and end positions were given randomly, as shown in Table 3. MuL-ACO was compared with ACO, MH-ACO, and MF-ACO in larger and more complex uneven environments.

**Table 3.** Simulation environment description.

| Map NO. | Name | Start Point | End Point |
|---|---|---|---|
| 1 | X-type | (0.5, 0.5) | (25.0, 23.0) |
| 2 | Z-type | (1.0, 24.0) | (25.0, 10.0) |
| 3 | Complex1 | (0.5, 0.5) | (26.0, 23.0) |
| 4 | Complex2 | (0.5, 0.5) | (29.0, 24.0) |
| 5 | Complex3 | (1.0, 0.5) | (29.0, 27.0) |
| 6 | Vortex | (28.0, 20.0) | (13.0, 15.0) |

Figure 9 shows the best trajectory planned by each method. It is obvious that the trajectory planned by the method proposed in this paper had fewer turns, good smoothness, and short length, while bypassing steep or concave terrain. In simple scenes, as shown in Figure 9c,d,e, the trajectory planned by MuL-ACO was smoother and bypassed steep or concave areas as much as possible. In more complex scenes (Map 1 and Map 6), the difference in the quality of the trajectories was more obvious. As shown in Figure 9a, the trajectories planned by ACO and MH-ACO extended along the edge of the obstacle resulting in multiple turns. Although the MF-ACO algorithm avoided local traps when planning trajectories, its trajectory height and smoothness were worse than MuL-ACO. Especially in Vortex map 6, as shown in Figure 9f, the traditional ACO failed to generate a trajectory from the starting point to the end point, and MH-ACO often fell into a local trap and could not escape. Although MF-ACO could work normally, the quality of the trajectory was not ideal. MuL-ACO had a good performance due to the improved heuristic function and adaptive pheromone adjustment strategy. Moreover, the trajectory optimization algorithm based on mutual learning is suitable for different maps. Therefore, the scheme proposed in this paper is not affected by changeable environments and can plan the trajectory with a good comprehensive quality.

As shown in Table 4, 30 experiments were performed on six different maps to obtain the specific average performance metric of the algorithms. Figure 10 shows the performance comparison of MuL-ACO and the other three algorithms on 6 maps, which more clearly highlights the advantages of the proposed scheme in terms of length, number of iterations, number of turns, and the height difference of trajectory. From the specific data and the line graph, it can be seen that the method proposed in this paper was significantly better than the other algorithms in the number of iterations and the number of turns, reducing them by about 66.7%~87.5% and 80%~94.7%. Moreover, compared with other algorithms, the trajectory length was reduced by about 6%~23.3%. Furthermore, the MuL-ACO scheme did not choose a steep route to reduce the wear of the mobile robot in uneven terrain. The height difference of the trajectory was reduced by 15.9%~68.2%. As shown in Table 4, in Maps 1, 2, 3, and 6, although the running time of MuL-ACO was not the shortest, the gap between the running time of MuL-ACO and the shortest running time was no more than 0.3 s. In some complex maps (Map4 and Map5), the running time of MuL-ACO was even about 0.1 s~0.8 s less than other algorithms, which provides the possibility for trajectory planning in dynamic environments.

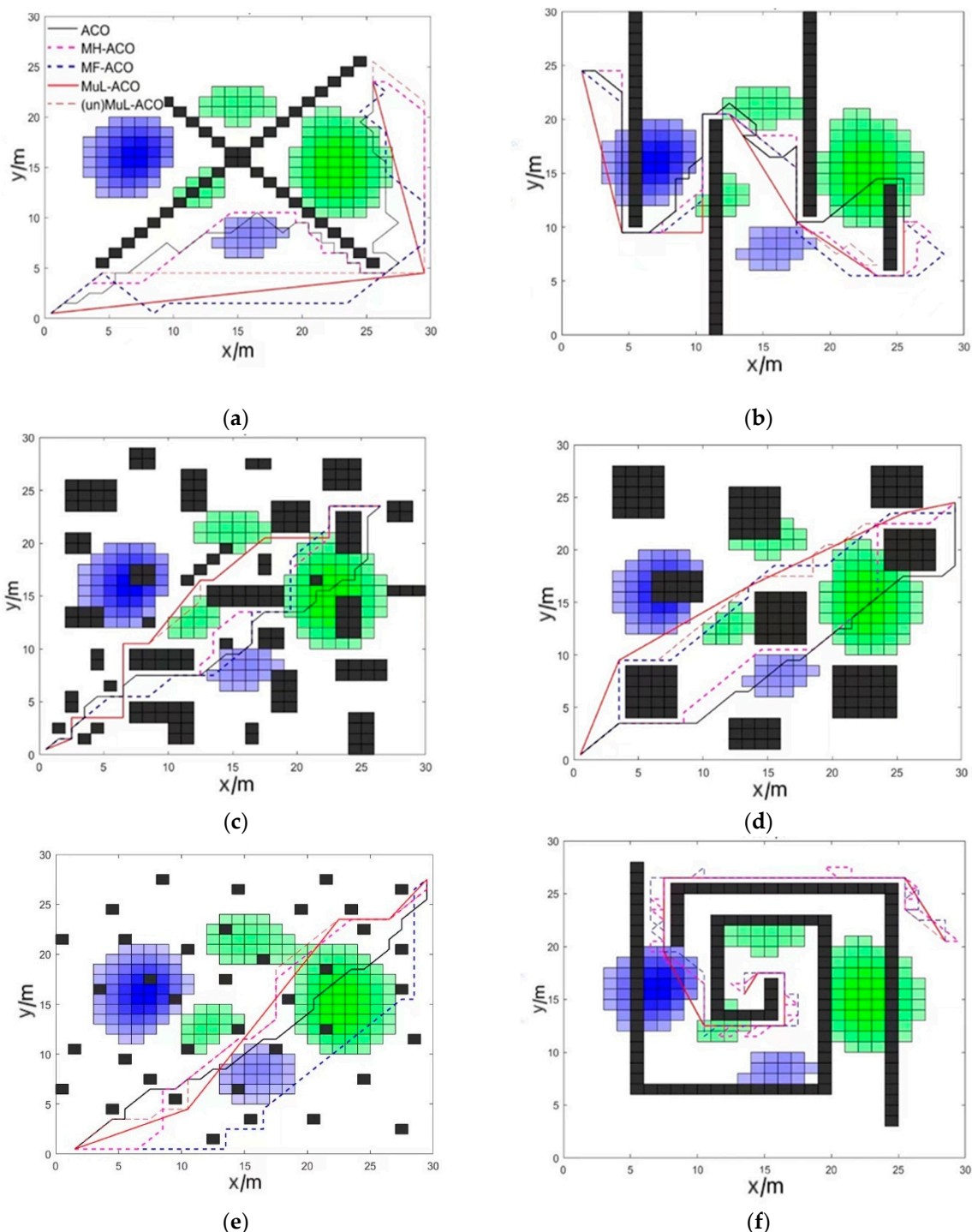

**Figure 9.** The trajectories generated by ACO, MH-ACO, MF-ACO, and MuL-ACO. (**a**) Map. NO.1: X-type Map; (**b**) Map. NO.2: Z-type map; (**c**) Map. NO.3: Complex1 map; (**d**) Map. NO.4: Complex2 map; (**e**) Map. NO.5: Complex3 map; (**f**) Map. NO.6: Vortex map.

**Table 4.** Simulation results of ACO, MH-ACO, MF-ACO, and MuL-ACO.

| Name | Map NO. | Length | Iterations | Turns | Height Difference | Time (s) |
|---|---|---|---|---|---|---|
| ACO | 1 | 60.80 | 40 | 38 | 7.59 | 2.02 |
| | 2 | 67.41 | 39 | 25 | 16.71 | 2.73 |
| | 3 | 41.2 | 38 | 22 | 7.36 | 1.41 |
| | 4 | 43.60 | 32 | 17 | 6.48 | 1.63 |
| | 5 | 43.0 | 30 | 14 | 7.45 | 1.71 |
| | 6 | —— | —— | —— | —— | —— |
| MH-ACO | 1 | 56.82 | 27 | 15 | 2.29 | 2.30 |
| | 2 | 66.30 | 21 | 13 | 9.31 | 2.53 |
| | 3 | 44.21 | 28 | 14 | 6.72 | 1.54 |
| | 4 | 43.40 | 25 | 8 | 5.10 | 1.72 |
| | 5 | 43.60 | 23 | 12 | 1.55 | 1.74 |
| | 6 | 78.40 | 19 | 31 | 8.15 | 2.57 |
| MF-ACO | 1 | 54.27 | 15 | 10 | 3.44 | 1.93 |
| | 2 | 74.10 | 12 | 14 | 9.98 | 2.87 |
| | 3 | 44.57 | 23 | 15 | 5.54 | 1.96 |
| | 4 | 43.41 | 18 | 9 | 3.33 | 2.28 |
| | 5 | 46.80 | 13 | 7 | 4.38 | 2.39 |
| | 6 | 73.62 | 16 | 22 | 6.93 | 2.54 |
| MuL-ACO | 1 | 46.38 | 6 | 2 | 1.51 | 2.11 |
| | 2 | 60.21 | 5 | 6 | 5.34 | 2.65 |
| | 3 | 42.88 | 5 | 9 | 2.18 | 1.61 |
| | 4 | 39.70 | 7 | 2 | 0.42 | 1.52 |
| | 5 | 43.10 | 8 | 3 | 1.56 | 1.55 |
| | 6 | 55.93 | 9 | 7 | 5.88 | 2.83 |

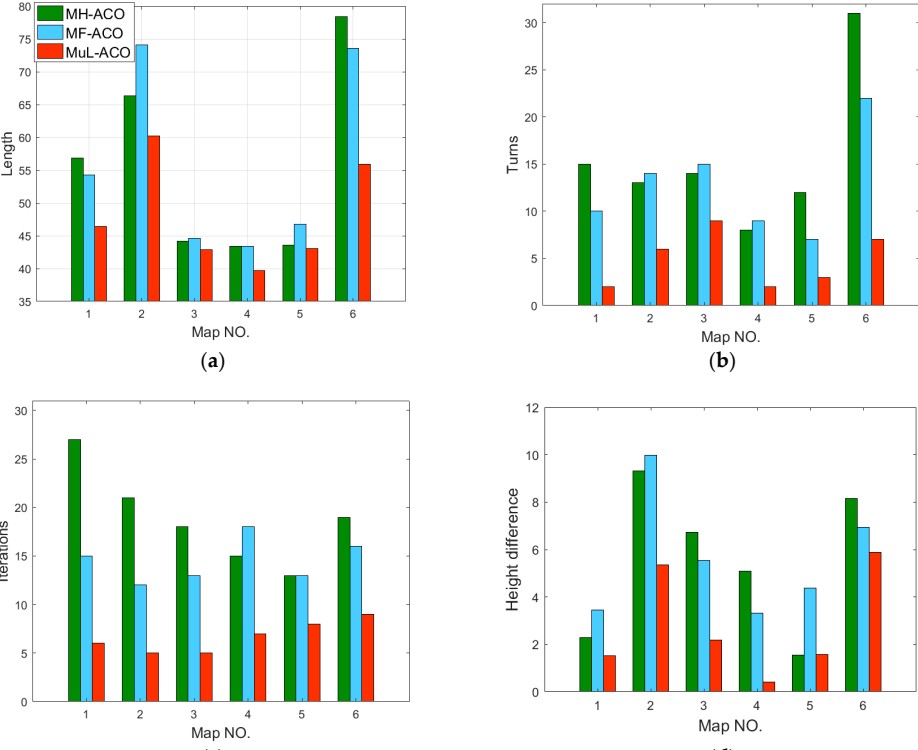

**Figure 10.** Comparison of trajectory performance. (**a**) length, (**b**) number of turns, (**c**) number of iterations, (**d**) the height difference of the trajectory.

### 6.3. Simulation Experiment C

In this group of experiments, the Complex2 map of simulation experiment B was selected. Moreover, different starting points and ending points were randomly set to diversify the experimental results. Figure 11 shows the optimal trajectories planned by ACO, MH-ACO, MF-ACO, and MuL-ACO. Table 5 summarizes the qualitative comparison of the performance of algorithms.

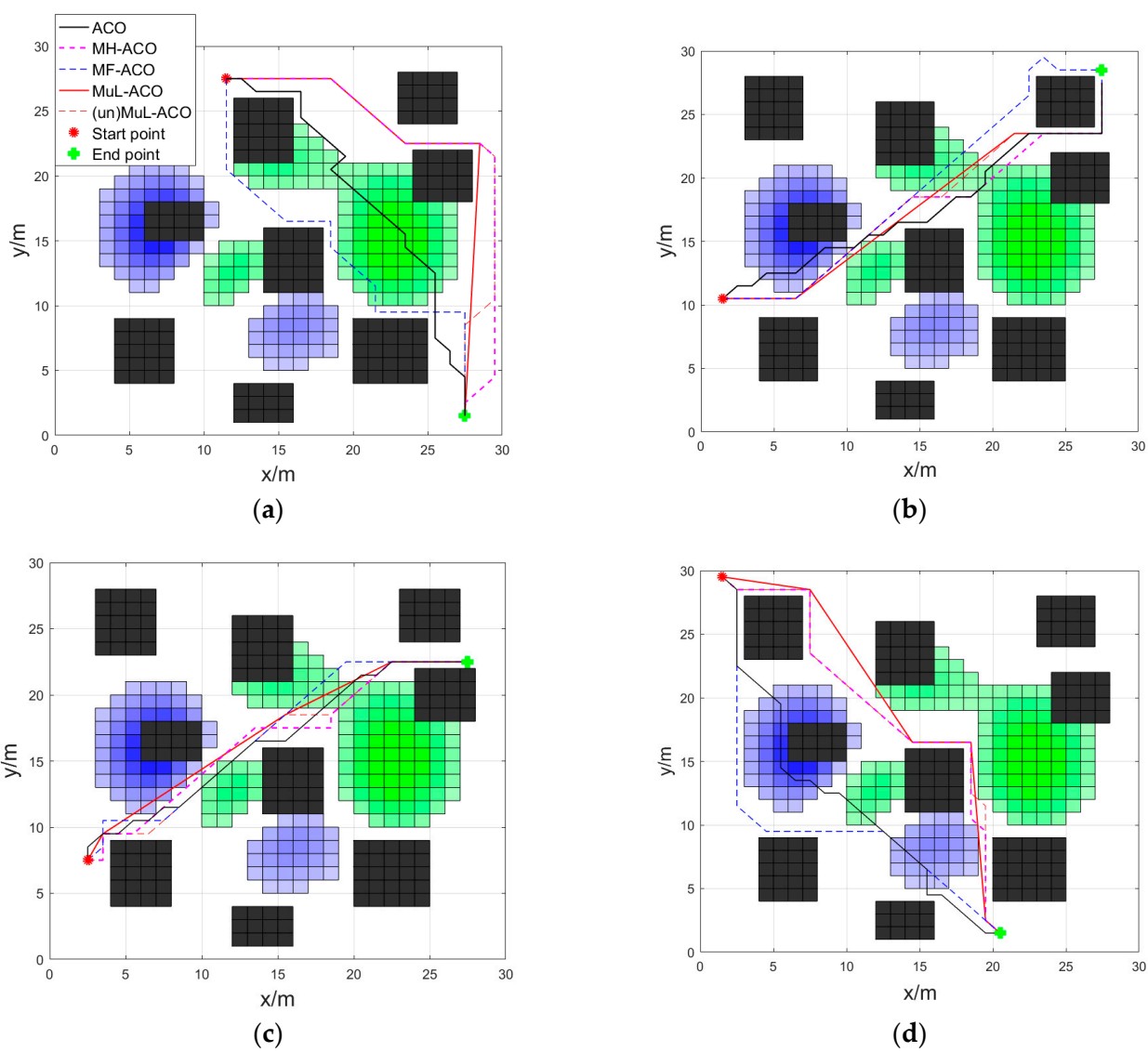

**Figure 11.** (**a**–**d**) The trajectories are generated by ACO, MH-ACO, MF-ACO, and MuL-ACO on the map with different starting points and ending points.

**Table 5.** Simulation results of ACO, MH-ACO, MF-ACO, and MuL-ACO on the map with different starting points and ending points.

| Name | Map (with Different Start and End Points) | Length | Iterations | Turns | Height Difference | Time (s) |
|---|---|---|---|---|---|---|
| ACO | a | 34.80 | 37 | 11 | 6.81 | 1.51 |
| | b | 35.80 | 25 | 16 | 7.44 | 1.40 |
| | c | 31.62 | 23 | 12 | 3.63 | 1.53 |
| | d | 38.60 | 15 | 21 | 11.02 | 1.52 |
| MH-ACO | a | 41.80 | 23 | 5 | 0.34 | 1.63 |
| | b | 35.20 | 17 | 5 | 1.49 | 1.53 |
| | c | 32.80 | 14 | 8 | 2.65 | 1.55 |
| | d | 41.00 | 11 | 8 | 0.38 | 1.66 |
| MF-ACO | a | 36.60 | 17 | 7 | 2.40 | 2.20 |
| | b | 36.20 | 13 | 6 | 2.07 | 2.10 |
| | c | 32.20 | 9 | 4 | 3.23 | 2.24 |
| | d | 40.20 | 10 | 4 | 4.33 | 2.13 |
| MuL-ACO | a | 40.09 | 11 | 3 | 0.39 | 1.38 |
| | b | 34.84 | 9 | 3 | 1.49 | 1.12 |
| | c | 30.89 | 10 | 3 | 3.04 | 1.24 |
| | d | 39.42 | 6 | 4 | 0.38 | 1.50 |

In Figure 11a,d, compared with other algorithms, MuL-ACO found a flatter and smoother trajectory for the robot. Although MH-ACO and MF-ACO also tended to find flat trajectories, the planned trajectories were not smooth enough. In Figure 11b,c, the trajectories planned by different algorithms were concentrated in the same area with height. Moreover, the cost of bypassing this height area was high. The hybrid scheme proposed in this paper did not completely bypass the height area but planned a shorter and smoother trajectory with a small height difference.

As shown in Table 5, the MuL-ACO hybrid scheme proposed in this paper had great advantages in smoothing performance. In terms of the height difference of the trajectories, the trajectories planned by MuL-ACO tended to bypass uneven areas, and the total height difference of the trajectories was lower than that of ACO, MH-ACO, and MF-ACO. Further, compared to other algorithms, MuL-ACO had the least number of iterations and the shortest running time. The hybrid scheme proposed in this paper runs stably and plans trajectories with a high comprehensive quality.

*6.4. Simulation Experiment D*

In this section, the feasibility of the proposed scheme in dynamical environments, which consisted of 2 dynamic obstacles represented by red blocks and other static obstacles represented by black blocks, was tested. The starting point of the mobile robot was (3.5, 2.5), and the end point was set to (28.5, 28.5). The dynamic obstacles moved vertically down and right at a rate of 2 m per time step and moved a total of 6 steps. Figure 12 shows the process of MuL-ACO generating the optimal trajectory. In the initial trajectory generation stage, the planned trajectory avoided steep regions and dynamic obstacles in the environment and contained several redundant trajectory nodes. In particular, it can be seen that the planned trajectory actively bypassed the vertical downward moving obstacles and adjusted the trajectory to reach the end point. In the trajectory optimization stage, the length and smoothness of the initial trajectory were minimized by a mutual learning-based trajectory optimization algorithm. The specific length and smoothness of the initial trajectory and the optimized trajectory are given in Table 6. Clearly, the proposed scheme can stably plan desired trajectories in dynamic environments.

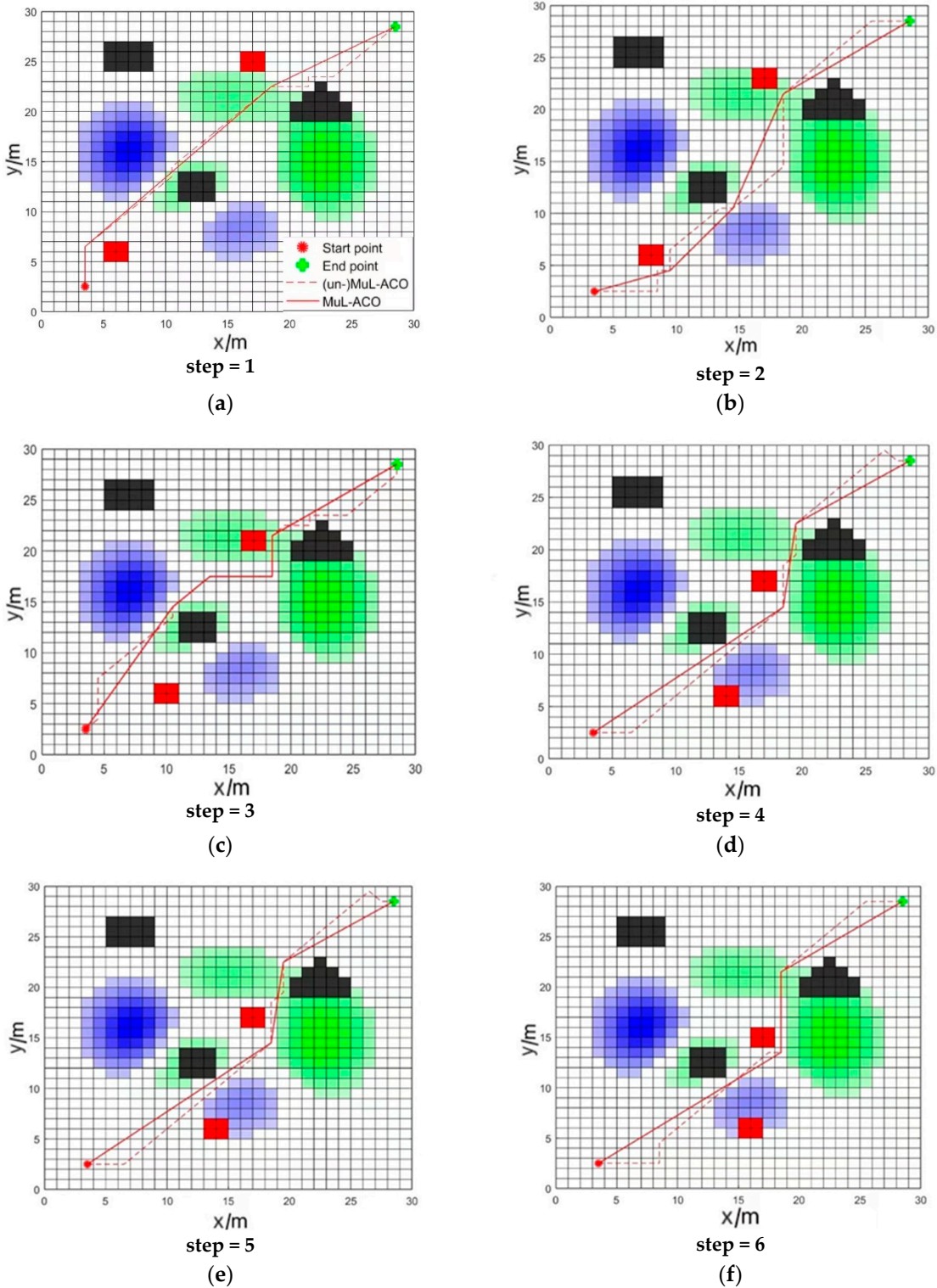

**Figure 12.** (**a**–**f**) Results of trajectory generation and optimization at different time steps (step = 1–6) in dynamic environments.

**Table 6.** Simulation results of (un)MH-ACO and MuL-ACO.

| Name | Map NO. | Length | Turns |
|---|---|---|---|
| (un)MuL-ACO | a | 41.6 | 7 |
| | b | 42.0 | 9 |
| | c | 40.8 | 11 |
| | d | 40.2 | 8 |
| | e | 40.8 | 7 |
| | f | 42.6 | 6 |
| MuL-ACO | a | 39.2 | 2 |
| | b | 39.1 | 3 |
| | c | 38.6 | 4 |
| | d | 38.3 | 3 |
| | e | 39.5 | 2 |
| | f | 39.3 | 2 |

## 7. Conclusions

Aiming at the trajectory generation and optimization of mobile robots in an uneven environment, a hybrid scheme using mutual learning and adaptive ant colony optimization (MuL-ACO) was proposed in this paper. The initial trajectory was generated by an improved adaptive ant colony algorithm, and then a mutual learning-based trajectory optimization algorithm completed the trajectory optimization. The comprehensive heuristic function and the adaptive method based on the improved temperature reduction function greatly improved the performance of the ant colony algorithm. Another advantage is that the proposed scheme had a clear division of labor to stably provide high-quality feasible solutions.

Experiments conducted in uneven environments of different scenes and sizes modeled by 2D-H maps showed that the trajectory planned by MuL-ACO was superior to the other five algorithms in terms of smoothness, height difference, length, and algorithm convergence. Especially in more complex and larger maps, the MuL-ACO scheme was more adaptable and could stably plan a trajectory with a higher comprehensive quality. In the future, the proposed hybrid method may be further optimized and could be applied to more complex scenarios, such as multi-agent scenarios with multiple objects and social interaction scenarios.

**Author Contributions:** Conceptualization: F.Q.; methodology: F.Q. and W.Y.; software: F.Q., W.Y., W.L. and K.X.; validation: W.Y. and C.L.; formal analysis, F.Q. and W.L.; investigation, W.L. and F.Q.; resources, F.Q. and W.Y.; data curation, F.Q. and W.Y.; writing—original draft, F.Q.; writing—review and editing, F.Q. and W.Y.; visualization, F.Q.; supervision, F.Q. and C.L.; project administration, F.Q. and W.Y.; funding acquisition, F.Q. and W.Y. All authors have read and agreed to the published version of the manuscript.

**Funding:** This research was funded by the National Natural Science Foundation (grant nos. 61602529 and 61672539). This work was also supported by the Hunan Key Laboratory of Intelligent Logistics Technology (2019TP1015) and Scientific Research Project of Hunan Education Department (No. 17C1650).

**Institutional Review Board Statement:** Not applicable.

**Informed Consent Statement:** Not applicable.

**Data Availability Statement:** Not applicable.

**Conflicts of Interest:** The authors declare no conflict of interest.

## Abbreviations

The following abbreviations are used in this manuscript:

| | |
|---|---|
| ACO | adaptive ant colony optimization |
| MuL-ACO | mutual learning and adaptive ant colony optimization |
| MH-ACO | multiple heuristics adaptive ant colony optimization |
| MF-ACO | multi-factor adaptive ant colony optimization |
| SA | simulated annealing |
| RGB-D | Red Green Blue-Depth |
| APF | artificial potential field method |
| GA | genetic algorithms |
| PSO | particle swarm optimization |
| GWO | grey wolf optimizer |
| IAACA | improved adaptive ant colony algorithm |
| MLTO | Mutual learning-based Trajectory Optimization Algorithm |
| SSA | sparrow search algorithm |

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
