# Peer review of "Trajectory Generation and Optimization Using the Mutual Learning and Adaptive Ant Colony Algorithm in Uneven Environments"

_applsci, doi:10.3390/app12094629_

Round 1

Reviewer 1 Report

This paper presents an algorithm based on ant colony to optimally find a collision-free trajectory from a start point to an end point within an uneven environment. First, the algorithm generates an initial trajectory, by adding factors (height, smoothness) in the heuristic function and updating the pheromone volatization factor to accelerate the convergence. Next, a mutual learning approach enables to optimize the length and smooth of the trajectory.

This paper is very-well written, the approach proposed to take into account the uneven environment is convincing and the numerous simulations enable to highlight the capabilities of the algorithm. The comparisons with other approaches are also well presented.

You could perhaps add a few words about future works.

Table 2 : Height (not Hight)

Reviewer 2 Report

In this paper, the authors consider a mutual learning and adaptive ant colony algorithm in an uneven environment. This algorithm reduces running time, trajectory length, height, and smoothness, which is verified by simulations. The topic is interesting. I have the following comments for the further improvement of the paper. 

  • In line 120. contribution 2 is too general and it consists of contributions 3 and 4. 

  • In Fig. 1 it is not clear how height characteristics in the 2D map affect the path planning i.e. when is the robot allowed to cross an area and when it has to get around an area. 

  • In equation (5) the authors mentioned that the heuristic function includes the length d, a smoothing function s, and a height function h, which are given in the following. However, the text explaining the equations is unclear, and s contains variable u which is not explained. Furthermore, i,j, and g are stated as girds instead of grid cells.

  • Equation (10) has the same condition for calculating T(i) in both cases.

  • In line 329, the abbreviation SSA is given, and not explained.

  • In line 341. all parameters of MuL-ACO, where it seems that the third parameter should be alpha. What is parameter C? Earlier in the paper C is defined as a vector, i.e. set of the coordinates in equation (11).

  • In line 371. there is a wrong reference to the table.

  • The text describing Table 4 should discuss the time of the trajectory planning. If trajectory planning is a few seconds, is it possible to use this algorithm for real-time applications and avoid collisions with dynamic obstacles? It is not clear how your algorithm works in the presence of dynamic obstacles.

  • The units of measurement are missing on the axes on all figures.

Reviewer 3 Report

This paper is about the creation of trajectories for mobile robots in a 2.5D environment. The algorithm is inspired by an ant colony-based algorithm. The paper also presents several optimization strategies, based on heuristics.

The goals of the optimization strategies are unclear.  In which size order would the elevations and hollows on the surface be? Which slopes would be permitted before being classified as an obstacle? What about thresholds, steps, etc. that could be passed by a mobile robot? Would these be part of the 2.5D map? Further, the resolution of the map and the elevations are not specified.

Which would be the optimization criteria? I.e., when would one route be preferred to the other?

The paper contains a number of acronyms. Please add a table of acronyms and abbreviations. Please define acronyms before use.

Lines 75ff: Please structure this paragraph.

There is a number of language issues that should be fixed before publication: Section name Line 430 should be "Conclusion". Also, Line 141: --> "Section 7 concludes the paper."

I find the article difficult to understand. It seems that the algorithms build on facts that are not explained in the paper. For example, it is unclear how the ant-colony algorithm works.

Lines 151ff / figures: The caption should contain more information, especially when there are several parts. In the text, it is referred to "the left/right of Figure 1". However, labelling these parts would be better.

Formulas page 5ff: some parts are in superscript in the text; why? The formulas are difficult to understand, and not all of them seem consistent. In Section 3.3, the term "temperature" is not introduced.

Line 210: what does "SA-based" mean?

Line 240ff: Please explain more how the benchmark functions are used, and how they are to be interpreted?

Figure 4: incomprehensible. No interpretation is given.

The trajectory optimization is unclear. What are the criteria for this optimization? Some parts of the line are very close to obstacles. Does this have any consequences for practical applications? Would some spline-based trajectories be more suitable?

Algorithm 1 is incomprehensible. It seems that the optimization criteria is the number of turns.

Line 298ff: This seems to be some heuristic algorithm.

In the figures for the experiment, the choice of parameters is not sufficiently explained. Further, an interpretation of the outcomes is not given. I see that the red line in Figure 8 seems to avoid heights or hollows.

Line 371: timeliness seems to be the wrong expression. Maybe: "run time"?

Table 3: How does the Map. NO. relate to Figure 9? In Figure 9, letters are used.

Figure 10: This type of diagram does not make sense. The graph would imply that a map "3.5" would be a possibility. Please use a bar diagram instead, where three bars would be placed adjacent to each other for each of the maps. Please also enlarge the fonts.

All Figures: Please use larger fonts for the legends.

The presented algorithms and heuristics are difficult to understand. The validity of the presented results is difficult to interpret. The evaluation method is not sufficient. Results are based on a few samples with similar start-/end-points. A study with more start- and end-points would describe the outcome better. A larger number of randomly assigned start-/endpoints would constitute a better experiment for evaluation.

Round 2

Reviewer 3 Report

The authors have addressed most of my concerns. I find the presented concept interesting.

Please consider the following comments for improvement:

  • In Section 2.2: The optimization considers several variables, such as length, height, and the number of turns. How would these optimization goals be weighted? I assume this is dependent on the application - however, this should be outlined. Further, instead of "height", the height difference per path length, slope, or similar would be a metric to measure how well the algorithm performs.
  • For the simulation experiment, it is still unclear how many different start- and end-points you use for the simulation. Do you use only the start- and end-points outlined in the figures for each map? Or, do you use all ranges along the borders (or a similar algorithm)? A number of representative paths through the maps, and presenting statistics about these results would be interesting to read about (if not already done). However, from your description, I cannot tell.
